# Adolescent Idiopathic Scoliosis in the Adult Patient: New Classification with a Treatment-Oriented Guideline

**DOI:** 10.3390/healthcare13192418

**Published:** 2025-09-24

**Authors:** Giovanni Viroli, Alberto Ruffilli, Matteo Traversari, Antonio Mazzotti, Marco Manzetti, Simone Ottavio Zielli, Alberto Arceri, Cesare Faldini

**Affiliations:** 1Department of Biomedical and Neuromotor Science—DIBINEM, University of Bologna, 40126 Bologna, Italy; giovanni.viroli@ior.it (G.V.); alberto.ruffilli@ior.it (A.R.); matteo.traversari@ior.it (M.T.); marco.manzetti@ior.it (M.M.); simoneottavio.zielli@ior.it (S.O.Z.); alberto.arceri@ior.it (A.A.); cesare.faldini@ior.it (C.F.); 21st Orthopaedic and Traumatologic Clinic, IRCCS Istituto Ortopedico Rizzoli, 40136 Bologna, Italy

**Keywords:** adolescent idiopathic scoliosis, adult idiopathic scoliosis, scoliosis classification, surgical strategy, spinal deformity, osteotomy, treatment algorithm, surgical planning

## Abstract

**Background/Objectives**: Adolescent Idiopathic Scoliosis persisting into adulthood (AAIS) presents progressive stiffening and degenerative changes that are not fully captured by existing classifications. This heterogeneity complicates clinical decision-making and surgical planning. The aim of this study was to propose a novel, treatment-oriented classification system for AAIS. **Methods**: A retrospective review was performed on patients with AAIS who underwent surgical correction between 2018 and 2022. Pre- and postoperative radiographs, CT scans, and MRI were analyzed to define curve characteristics and evaluate surgical outcomes. Subgroups were identified according to age and deformity features, and corresponding surgical strategies were outlined. **Results**: AAIS was stratified into Young Adult Idiopathic Scoliosis (YAdIS, 19–30 years) and Adult Idiopathic Scoliosis (AdIS, >30 years). YAdIS was divided into mild, flexible curves (YAdIS 1) and severe/stiff curves (YAdIS 2). AdIS was classified into three categories: AdIS 1 (isolated coronal deformity), AdIS 2 (combined coronal and sagittal deformity), and AdIS 3 (revision cases). Within AdIS 1, additional refinement by age (30–45, 45–60, >60 years) reflected increasing stiffness and degenerative changes. Tailored surgical strategies included selective fusions, posterior releases, high-density constructs, three-column osteotomies, and combined anterior–posterior approaches, depending on curve type and age group. **Conclusions**: This classification provides a comprehensive, treatment-oriented framework to support surgical decision-making in AAIS, enabling optimized planning and improved outcomes for adult patients with scoliosis of adolescent onset.

## 1. Introduction

Adolescent Idiopathic Scoliosis in the Adult patient (AAIS) is defined as an Adolescent Idiopathic Scoliosis (AIS) curve which is observed after the age of 18. AAIS curves have typically exhausted the rapid active aggravation of AIS during the pubertal growth spurt; however, they are characterized by a slow passive aggravation throughout life that becomes increasingly significant with advancing age. Furthermore, the altered biomechanical environment created by the deformity leads initially to a progressive retraction of the ligamentous structures, loss of rib cage flexibility, and ultimately to the onset of degenerative changes. This causes a progressive stiffening of both structural and non-structural curves, resulting in rigid main curves and in non-structural compensatory curves becoming structural in the adult patient. This entire process is directly proportional to the severity of the curve at the end of growth and to the patient’s age, resulting in an extremely wide spectrum of heterogeneous AAIS curves. In this view, a comprehensive classification system of AAIS, able to capture its complex pathological spectrum, is still missing. The Aebi classification of adult scoliosis [1] originally differentiated the nosological entities of de novo adult scoliosis and AAIS, but did not stratify the spectrum of AAIS, nor did it provide specific indications of treatment strategies. Moreover, it focused more on thoracolumbar–lumbar curves than on thoracic curves. Recently, Lin et al. [2] proposed a new AAIS classification system as an extension of the Lenke classification of AIS. Similarly to the original Lenke AIS classification, it serves as a guide for choosing fusion levels in AAIS curves; however, it fails to stratify the wide spectrum of different AAIS curves and to consequently give specific correction strategy recommendations. Therefore, the purpose of this work is to critically analyze the broad spectrum of AAIS curves, identifying subgroups of patients that are homogeneous for deformity characteristics, thus providing a treatment algorithm regarding correction strategies.

## 2. Materials and Methods

The authors retrieved a collection of AAIS cases, consecutively treated among their department from 2018 to 2022. A total of 380 patients were enrolled. Inclusion criteria were as follows: adult age (>18 years old), presence of scoliotic deformity of more than 40°, and idiopathic etiology of the deformity. Exclusion criteria were as follows: presence of scoliotic deformity less than 40° and different etiology of the deformity (degenerative or secondary). A total of 131 patients met the inclusion criteria (101 female and 30 male). The mean age was 44 years old. A retrospective, descriptive review of the characteristics of the deformity was performed by two senior authors. In particular, a careful evaluation of pre-operative and post-operative long-standing spinal X-rays, CT scans, and MRI was performed. Cases representing subgroups of AAIS curves were selected. The relevant variables influencing surgical planning and their impact on surgical outcomes in terms of triplanar correction were identified. In particular, categories were identified and refined based on expert clinical observation of key variables such as patient age, curve characteristics (severity, flexibility), and the presence of degenerative changes. Subsequently, for each subcategory of AAIS, the most appropriate indications regarding surgical correction strategies were assigned.

## 3. Results

First, the broad AAIS spectrum was subdivided using an age criterion. In fact, as previously stated, age plays a crucial role in the progression of AIS curves after skeletal maturity, and it is a critical factor to consider when planning the surgical correction of a curve. Therefore, adopting a previously described cut-off [3,4,5,6], we identified two separate entities: Young Adult Idiopathic Scoliosis (YAdIS; age 19–30) and Adult Idiopathic Scoliosis (AdIS; age > 30). This age-based distinction is based on the significant changes in curve flexibility and the onset of degenerative processes that occur around the age of 30, which directly influence surgical planning and outcome. Then, each of these two entities should be stratified according to the specific curves’ features.

### 3.1. Young Adult Idiopathic Scoliosis (YAdIS—19–30 Years Old)

#### 3.1.1. YAdIS 1

These were AIS curves that lay in the “gray zone” (40–50°) according to the SRS guidelines [7] and that were conservatively treated during their adolescent years. However, these patients often seek surgical correction of their deformity during their early adulthood, mainly for cosmetic concerns, rather than pain [8]. Given their mild deformity and their relatively young age, the pathoanatomical features of YAdIS curves do not differ that much compared to AIS curves. In fact, typically, these curves still retain a certain degree of flexibility both in the main curve and in the compensatory curves, which have generally not yet become structural. Considering that, in YAdIS 1 patients, surgical planning, the fusion area, and corrective strategies are comparable to the ones adopted for AIS patients, with similar expected correction outcomes (Figure 1).

In particular, in terms of fusion area, Lenke rules can still be followed [9], with the exception of rare cases that already show early intervertebral degeneration, in which the fusion area needs to be extended to include the degenerated units and avoid junctional disease (Figure 2).

Regarding correction strategies, the need for posterior releases should be limited to hypokyphotic or hyperkyphotic curves. Conversely, on the coronal and axial planes, an optimal correction is generally possible without the need for posterior releases.

#### 3.1.2. YAdIS 2

These were AIS curves that were already amenable to surgical correction in the adolescent years according to the SRS guidelines [7] but did not receive surgery during adolescence. The reasons why AIS patients and families choose to delay surgery can be multiple: concerns of conflicts with school or sports activities, as well as concerns about the influence of surgery on residual growth. Given their significant deformity, despite their relatively young age, these patients already show important pathoanatomical changes, with an increased stiffness both in the main and in the compensatory curves. This has important implications for surgical strategy. In particular, stiffer compensatory curves often become structural and should be included in the fusion area. On the other hand, when compensatory curves are still non-structural and do not lead to an extension in the fusion area, their increased stiffness raises the risk of iatrogenic coronal decompensation after selective fusion, due to their lower spontaneous correction rate [5].

Regarding the main curves, their increased stiffness requires us to adopt specific correction strategies, stratified by the severity and the stiffness of the curves, in order to have an optimal triplanar correction:-For curves < 90° OR ≥90° and with a flexibility index > 15% (YADIS 2A), a combination of strategies with an all-posterior approach, already published by the authors under the acronym HiPoAD (High-Density Pedicle Screws, Ponte Osteotomies, Asymmetric Rods Contouring, Direct Vertebral Rotation) [10], may be sufficient to address the deformity. In particular, high-density constructs are preferrable in order to dissipate the corrective forces on every level and decrease the pull-out risks. Then, an aggressive posterior release based on multiple asymmetric Ponte osteotomies is of paramount importance, in order to allow an optimal deroto-translation over two asymmetrically contoured rods (Figure 3) [10,11].

-For curves ≥ 90° AND with a flexibility index < 15% (YADIS 2B), the HiPoAD technique, even though it proved to be effective for this kind of curve when addressed during adolescent years [12], may not be powerful enough for YAdIS 2 patients. These cases should be addressed with a three-columnar approach. One strategy could be to adopt three-column osteotomies like VCR. A possible alternative, in order to avoid the risks of a three-column osteotomy, is to perform a three-columnar release through a combined approach: an anterior thoracoscopic release (wide resection of anterior longitudinal ligament and multiple periapical discectomies), followed by a posterior column release based on multiple Ponte Osteotomies and a posterior correction (VT-HiPoAD, Videothoracoscopic release—HiPoAD) (Figure 4).

### 3.2. Adult Idiopathic Scoliosis (AdIS—>30 Years Old)

#### 3.2.1. AdIS 1

These were AIS curves that lay in the “gray zone” (40–50°) during their adolescence, were conservatively treated during their adolescence and early adulthood years, but then presented for surgical correction after 30 years of age. In other words, they represent the progression of YAdIS 1 curves, with crucial differences between the two, determined by the passage of time. First, the result of the passive progression of the curves is already seen, with curves almost invariably exceeding 60–70°. Moreover, a significant stiffness both in the main and in the compensatory curves is seen. Compensatory curves almost invariably become structural, and the prevalence of true Lenke 1 and 5 patterns gradually decreases in the AdIS 1 group, in favor of Lenke 2, 3, 4, and 6 patterns. Additionally, disc degeneration in the lumbar compensatory curves becomes increasingly more prevalent with age. Therefore, the fusion area becomes significantly more extensive in AdIS 1 compared to the YAdIS 1 group, and selective fusions become less frequent. An additional factor of paramount importance that should be considered in surgical planning is the lumbosacral fractional curve. When the fractional curve becomes structural and/or degeneration is present in the lumbosacral junction, the fusion area should include the lumbosacral curve, extending to the sacrum and/or pelvis as suggested by Lin et al. [2]. Regarding the main curve correction, its more severe amplitude due to passive progression and its increased stiffness require an additional stratification based on the age of patients:-***AdIS 1A:* 30–45 years old.** In this subset of patients, combined corrective strategies based on posterior releases (Hi-PoAD [10]) can still overcome the moderate anterior constraint to derotation and translation, resulting in an effective triplanar correction (Figure 5).

-***AdIS 1B:* 45–60 years old.** These patients usually have a stiffer main curve, with the not infrequent presence of anterior column osteophytes, especially at the concave side. In this setting, combined corrective strategies based on posterior releases (Hi-PoAD [10]) can still achieve an optimal translation, but often, derotation potential is limited, with a possible residual of a partial rib hump (Figure 6).

-***AdIS 1C:* >60 years old.** This is infrequent, since patients in this age group more commonly tend to lie in the AdIS 2 group. This group is burdened by an increased stiffness due to an even more frequent anterior column spontaneous fusion, resulting in a less powerful correction both in terms of translation and derotation, with a consequent smaller correction rate and more prominent residual rib hump. However, it must be considered that these patients more commonly present to surgeons because of pain rather than cosmetic concerns. In this view, pain generators should be carefully identified and adequately treated, performing foraminal and/or central decompressions and discectomies where appropriate (Figure 7).

#### 3.2.2. AdIS 2

These patients have a combined deformity, with a concomitant degenerative sagittal plane deformity that develops alongside a pre-existent adolescent coronal deformity. The most common sagittal deformities are thoracolumbar junction kyphosis (Figure 8) and flat-back syndrome [13] (Figure 9). These sagittal deformities tend to occur earlier in the AdIS population than in the general population, due to the altered biomechanical environment that exposes the scoliotic spine to an increased risk of degenerative cascade. Moreover, pregnancy, besides its well-known coronal progression risk in AIS curves [13], has a kyphotic effect [14,15] and can act as a trigger to an even earlier development of sagittal deformities in AdIS patients, particularly in the case of thoracolumbar/lumbar curves (Figure 8).

AdIS 2 patients require a case-by-case surgical strategy. First of all, sagittal deformity determines pain and disability in these patients [8], thus representing more often the main clinical issue of these patients, compared to coronal deformity. In this regard, once surgery allows us to achieve a proper pain-free sagittal alignment and a global coronal balance, a residual coronal curve is well tolerated by these patients. Multiple posterior column osteotomies usually allow an optimal correction when sagittal deformities retain a certain flexibility and/or are not sharp. Conversely, when sagittal deformities are stiff and/or sharp and when a significant global sagittal imbalance is present, three-column osteotomies are indicated (Figure 9).

#### 3.2.3. AdIS 3

This group comprises curves that have already been surgically treated in the past and seek further surgery due to one or more of the following: not tolerated residual coronal deformity, sagittal imbalance due to distal or proximal junctional kyphosis, iatrogenic flat-back or the kyphotic effect of the previous instrumentation, and pseudoarthrosis. The presence of the fusion mass significantly poses unique challenges during the revision surgery on several levels. First, freehand pedicle screw positioning in the fusion mass can be demanding due to the altered anatomic landmarks: robot-assisted placement [16], navigation [17], or patient-specific 3D-printed guides [18] (Figure 10) may offer a very helpful aid in the identification of the correct entry point and trajectory of the screw. Alternatively, the use of fusion mass screws has been introduced as a salvage procedure for these cases, but with a limited number of reported cases [19,20]. Secondly, the fusion mass requires almost invariably three-column osteotomies in order to correct any residual deformity on the coronal plane or to correct any sagittal imbalance (Figure 10).

## 4. Discussion

To identify and specifically address each subgroup of the broad AAIS spectrum is a real challenge that still remains unsolved to this day. In the present paper, a new classification for AAIS curves is proposed, with a stratification of patients according to deformity characteristics and with a consequent treatment-oriented guideline. We accordingly present here a flowchart (Figure 11).

The first distinction was made between YAdIS (19–30 years old) and AdIS (>30 years old) patients. The background for this decision comes from multiple studies that previously defined the YAdIS category. However, contrasting results have emerged from match cohort studies between YAdIS and AIS patients [3,5,6,21], from which the need to have a further stratification of YAdIS patients became clear. In fact, in a study by Chan et al. [21], YAdIS and AIS patients had a similar preoperative amount of curve flexibility, as well as comparable postoperative outcomes, in terms of correction rate, blood loss, hospital stay, and fused levels. This suggests the existence of a subset of YAdIS patients (YAdIS 1) that share similar deformity features with AIS patients, resulting in similar correction strategies and predicted outcomes. On the other hand, a subset of YAdIS patients present with important curves that require specific correction strategies (YAdIS 2). In this subgroup of patients, when facing curves from 50 to 100°, multilevel posterior releases may still be sufficient. However, compared to AIS patients, there is a higher prevalence of severe curves among YAdIS patients, as detected by Kurra et al. [6]. Therefore, in his cohort of YAdIS patients, there was a higher rate of combined approaches and of tri-column osteotomies. Our classification is in agreement with this view, and tri-column osteotomies or anterior + posterior releases should be considered when addressing severe (>100°) YAdIS 2 curves, with some differences. Three-column osteotomies like VCR have a segmental action, focused just on the apical level, while anterior + posterior releases spread their correction on multiple apical–periapical levels. Considering that idiopathic scoliosis is generally characterized by relatively large radius curves spread on more than five levels, a more distributed correction provided by anterior + posterior releases may offer some advantages. Moreover, an often-neglected issue with these patients is that they usually have a mismatch between trunk height and leg length due to the severity of the spinal deformity. The use of three-column osteotomies like VCR, which are shortening procedures, may therefore result in failure to restore a proper trunk height/leg length ratio. In this view, anterior + posterior releases may offer an additional cosmetic advantage. Given that, in the authors’ view, three-column osteotomies like VCR remain a fundamental weapon, but with limited indications such as the following: extremely sharp deformities, often with a kyphotic component and associated myelopathy. These features are more typical of congenital or secondary scoliosis rather than idiopathic scoliosis, and, therefore, VT-HiPoAD is our treatment of choice.

After the age of 30, the characteristics of the curves, even of the mild ones, change drastically. Firstly, compensatory curves as well as the fractional lumbosacral curve almost invariably become structural. As demonstrated by Lonner et al. in their match cohort study, in fact, fusion can become, on average, 3.5 levels longer in AdIS patients, and fusion to the pelvis can be required in 36% of them [22]. Moreover, it must be considered that curves lose their flexibility dramatically and quickly during adulthood [23]. Deviren et al. [24] reported a structural thoracolumbar/lumbar curve average flexibility decrease of 5% with every 10-year increase in age. However, it must be considered that the study included patients with a wide age range (13–78 years, average 41 years). Therefore, we can assume that, considering only adult patients, the flexibility decrease would be even greater than 5% every 10 years, with an exponential trend. It becomes clear from this that further age-based stratification of AdIS1 patients in groups A, B, and C was necessary in order to account for the decreased corrective potential of the main curve achievable through posterior releases, particularly on the axial plane, due to its increased age-related stiffness. Patients should be carefully informed of that, although it must be considered that, as stated in previous reports [8,25], older patients usually refer to surgery for pain and functioning impairment, rather than cosmetic concerns. AdIS 2 patients share traits with adult degenerative deformity patients, both from a pathoanatomical and clinical point of view. In fact, AdIS 2 represents a group of patients with an idiopathic coronal deformity in which the degenerative cascade results in the development of a sagittal deformity, with the related pain and disability constituting the main issues for these patients. In this view, it is crucial to prevent the degenerative cascade in AdIS patients, as far as possible, through a careful follow-up of patients, an accurate assessment of bone mineral density, excellent maintenance of muscle tone, and weight control, in order to avoid or delay as much as possible the development of sagittal deformities [26,27].

AdIS 3 patients are a new challenge that spinal deformity surgeons are facing nowadays more and more often. On the one hand, surgeons are facing the limitations of previous generations of spinal instrumentations. For example, the Harrington rod had a typical kyphotic effect that can result in sagittal imbalance due to flat-back syndrome or junctional disease and had limited coronal correction potential compared to modern instrumentations [28]. On the other hand, early implanted modern spinal instrumentations were also not free from unsatisfactory results requiring revisions. Treating this group of patients poses unique challenges, and unfortunately, only limited reports are available in the modern literature. Louie et al. [28], in their multicenter study of revision surgery patients after Harrington rod instrumentation, reported that 35% of their patients underwent combined anterior–posterior fusions, 65.9% underwent an osteotomy (41.5% posterior column osteotomy, 24.4% PSO), with a 61% complication rate.

It is crucial to present patients with the risk–benefit profile of various procedures that may be proposed. It is important to consider that a patient with adult idiopathic scoliosis (AIS) is subject to a significantly higher rate of complications with advancing age, as has been widely demonstrated in the recent literature. Specifically, it has been shown that patients with AIS have greater blood loss, longer operative times [29,30], and higher complication rates, particularly in terms of intraoperative complications, including neurological ones, and complications during the hospital stay [29,31].

However, increased evidence [29,30] suggests that adult patients have a greater expectation of improvement in self-image, despite the increased surgical risk. This can be due to two main reasons: a lower starting point and specific factors influencing their self-perception. Before the surgery, older adult patients had the absolute lowest self-image scores compared to adolescents and young adults. This means they had a much larger room for improvement: while they achieved similar post-operative self-image scores as younger patients, their journey of improvement was the most significant. Additionally, the factors influencing self-perception vary with the patient’s age. For adolescents and young adults, self-perception is linked to coronal alignment issues and mental health diagnoses, whereas the situation is different for older adults. In this age group, the main predictor of worse post-operative self-image was a greater residual pelvic tilt (PT), which is part of sagittal alignment. This suggests that for them, satisfaction is not just about curve correction; it is closely tied to specific aspects of overall alignment, particularly the sagittal plane [29].

The presented classification then emphasizes the possibility that a patient, following the natural history of the pathology, which is typically progressive throughout life, can move from one subcategory to another during a lifespan. Indeed, a recent study [32] showed that even moderate curves (30–40°) at skeletal maturity exhibit lifelong progression characteristics, with a rate of 0.5° per year. In this view, the classification implies that all adult idiopathic scoliotic curves, even when relatively mild, require constant and periodic lifelong follow-up to detect any progression.

Classifications allow us to identify categories of patients with different causes, behaviors, needs, or prognosis. The present classification of Adolescent Idiopathic Scoliosis in the Adult patient allows us to identify categories of patients with different characteristics, providing a treatment-oriented guide for surgeons in order to help them achieve better clinical outcomes for every patient. However, this study does not come without limitations. We are aware that the non-experimental nature of this work is its primary limitation. Rather than presenting a systematic analysis, the manuscript relies mainly on descriptive case series and hypotheses derived from the authors’ experience. The lack of statistical validation or objective assessment that supports the proposed system limits the reliability and generalizability of the classification itself.

However, we believe that this approach is crucial for an initial proposal of a complex, treatment-oriented classification system. The criteria for each category (e.g., age cutoffs, curve stiffness, presence of degenerative sagittal deformity) are based on distinct clinical features that necessitate different surgical approaches, and the presented cases are representative examples to illustrate each proposed category, making the system clinically understandable and applicable.

Secondly, it is difficult to produce a treatment-oriented approach based solely on experimental data due to the complexity of the problem and the numerous variations discovered.

Additionally, our analysis is based on a relatively small sample base from a single institutional experience, which may introduce a potential for selection bias and limit the generalizability of our findings. Moreover, as an initial proposal, this manuscript does not include long-term follow-up data to assess the durability of surgical outcomes or the long-term impact on patients’ quality of life.

This article therefore condenses the authors’ extensive knowledge, which will then be validated as a whole system in future studies.

In fact, the manuscript serves as a foundational proposal, but further research is needed to provide future validation of the classification, with inter- and intra-observer reliability studies and analyses correlating the classification to clinical outcomes and complication rates.

## 5. Conclusions

Adolescent Idiopathic Scoliosis in the Adult patient (AAIS) represents a complex spectrum of deformities with different subgroups of patients for deformity characteristics, which should be specifically identified in order to give the most optimized surgical solution to each patient.

## Figures and Tables

**Figure 1 healthcare-13-02418-f001:**
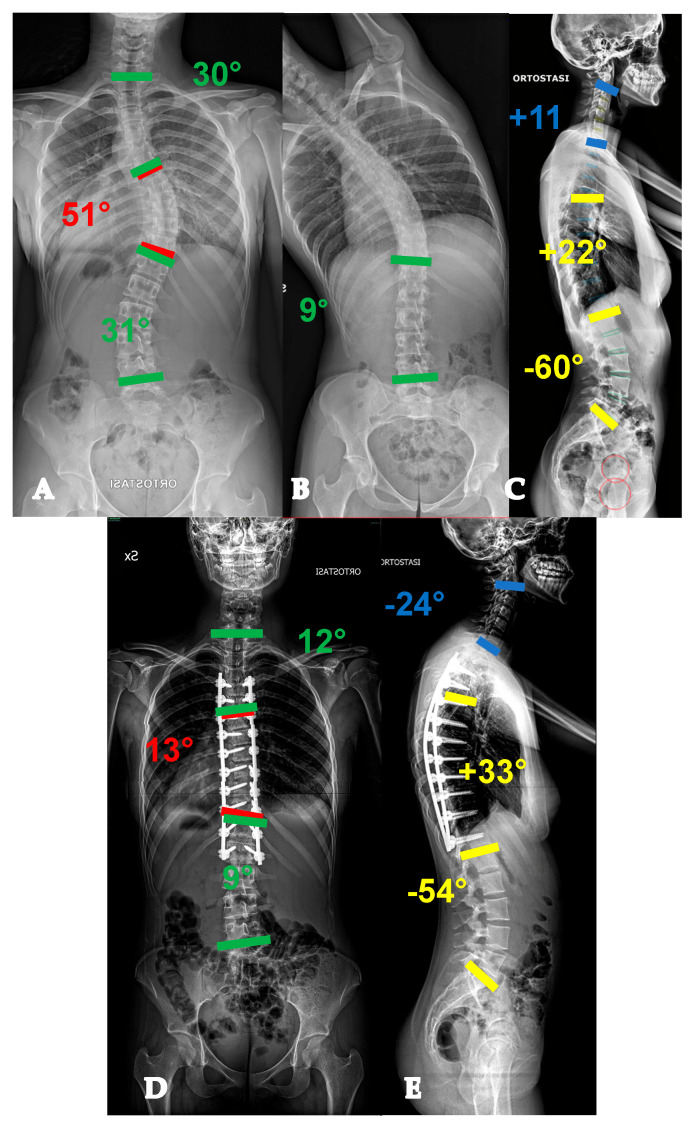
Twenty-six-year-old lady with a YAdIS 1 curve. She has a main thoracic curve with a non-structural lumbar curve (Lenke 1 pattern) (**A**,**B**). On the sagittal plane (**C**), a mild hypokyphosis can be seen with a compensatory cervical kyphosis. We opted for a T3-L1 selective fusion with strategic Ponte Osteotomies in order to restore an ideal sagittal alignment with thoracic kyphosis and cervical lordosis restoration (**D**,**E**).

**Figure 2 healthcare-13-02418-f002:**
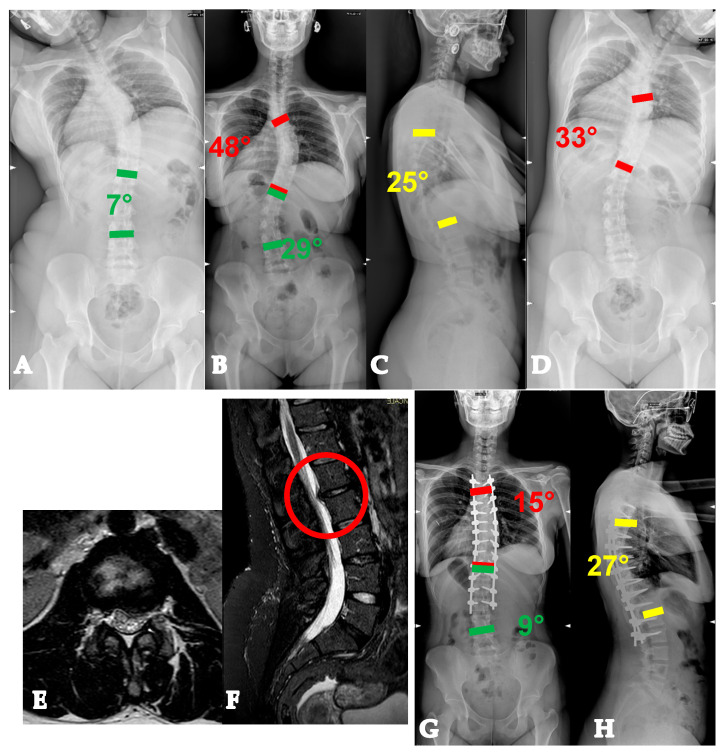
Twenty-two-year-old lady with a YAdIS 1 curve. She has a main thoracic curve with a non-structural lumbar curve (Lenke 1 pattern) (**A**–**D**). However, on the MRI, she already had significative L1-L2 disc degeneration (**E**,**F**), which was symptomatic. We therefore opted for a T3-L2 non-selective fusion (**G**,**H**).

**Figure 3 healthcare-13-02418-f003:**
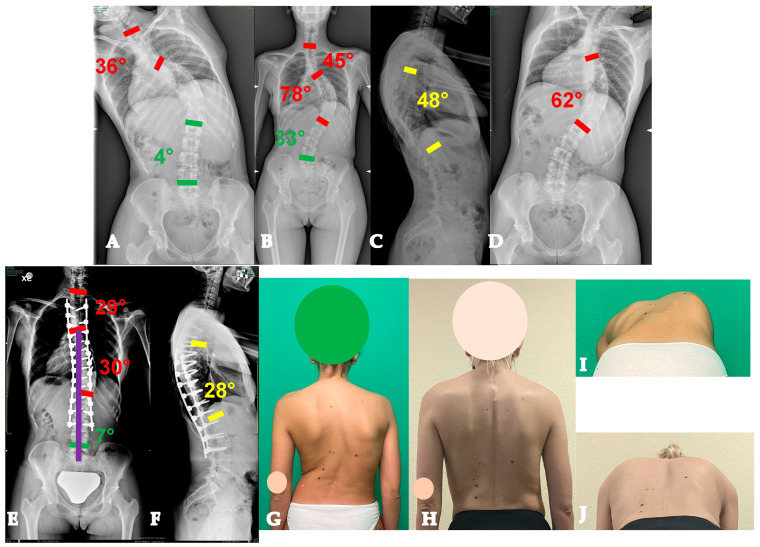
Twenty-two-year-old lady with a YAdIS 2A curve. She has a main thoracic curve with a right L4 tilt and a structural proximal thoracic curve (Lenke 2AR pattern) (**A**–**D**). The main curve is below 90°, so an all-posterior approach via the HiPoAD technique was adopted. This ultimately resulted in an optimal correction both radiographically (**E**,**F**) and clinically (**G**–**J**).

**Figure 4 healthcare-13-02418-f004:**
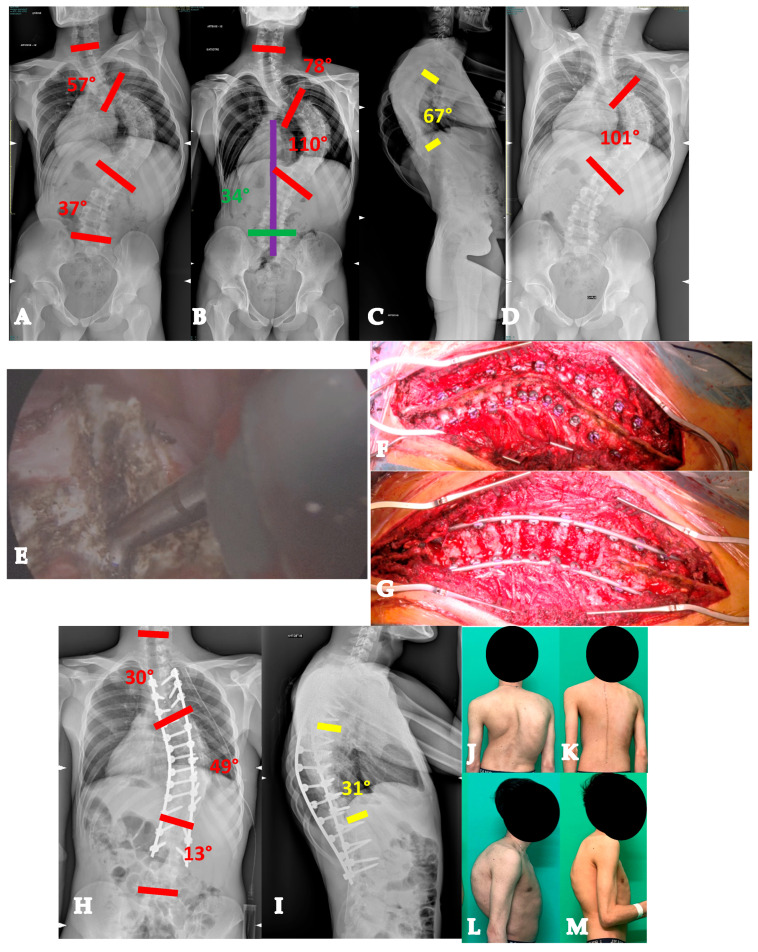
Twenty-one-year-old man with a YAdIS 2B curve. He has a severe main thoracic curve with a structural proximal thoracic curve and lumbar curve (Lenke 4 pattern) (**A–D**). The main thoracic 110° with a severe stiffness (**B**,**D**). Therefore, a VT-HiPoAD strategy was adopted: first, thoracoscopic discectomies were performed (**E**), followed by a posterior wide release and fusion (**F** before maneuver, **G** after maneuver). A good radiographic (**H**,**I**) and clinical (**J–M**) result was achieved.

**Figure 5 healthcare-13-02418-f005:**
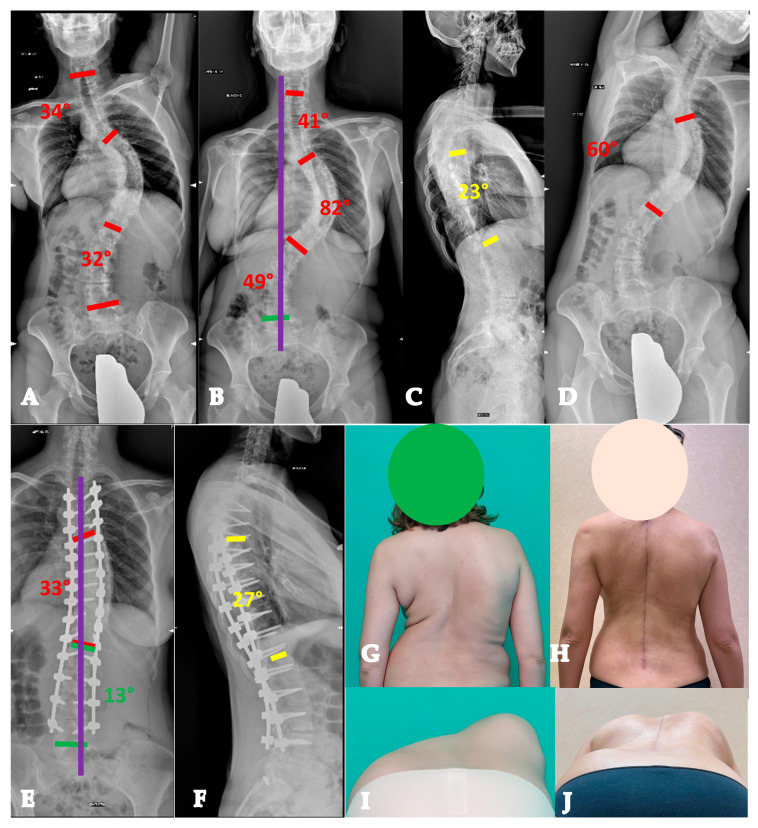
Forty-three-year-old woman with an AdIS 1A curve. Despite her severe and relatively stiff deformity (**A**–**D**), a T3-L4 fusion with the Hi-PoAD technique (**E**,**F**) allowed us to achieve a good clinical and radiographical result (**G–J**).

**Figure 6 healthcare-13-02418-f006:**
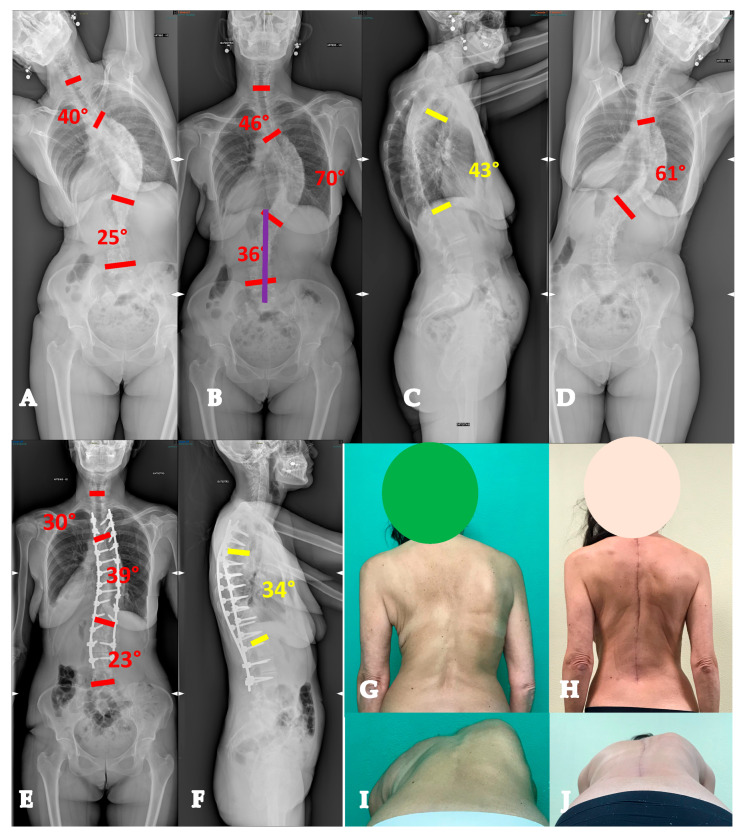
Fifty-five-year-old woman with an AdIS 1B curve. She had a very stiff curve (**A**–**D**), so the T3-L3 (**E**,**F**) fusion with the Hi-PoAD technique allowed us to achieve a good result; however, a residual hump can be seen (**G**–**J**).

**Figure 7 healthcare-13-02418-f007:**
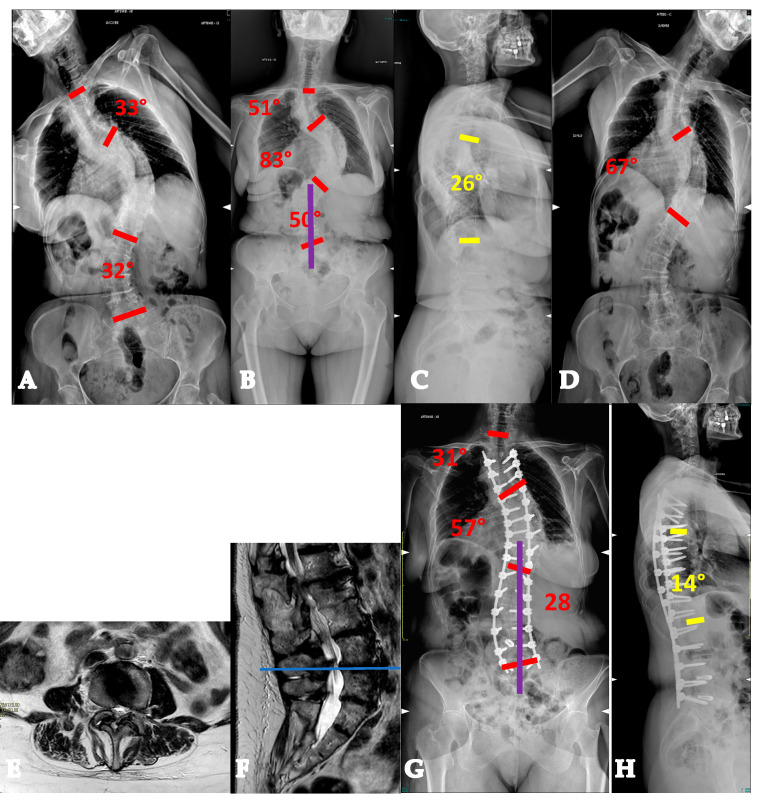
Sixty-one-year-old woman with an AdIS 1C curve (**A**–**D**). She had low back pain and right leg pain. The MRI (**E**,**F**) showed L4-L5 foraminal and central stenosis; therefore, direct decompression was performed alongside the posterior deformity correction (**G**,**H**).

**Figure 8 healthcare-13-02418-f008:**
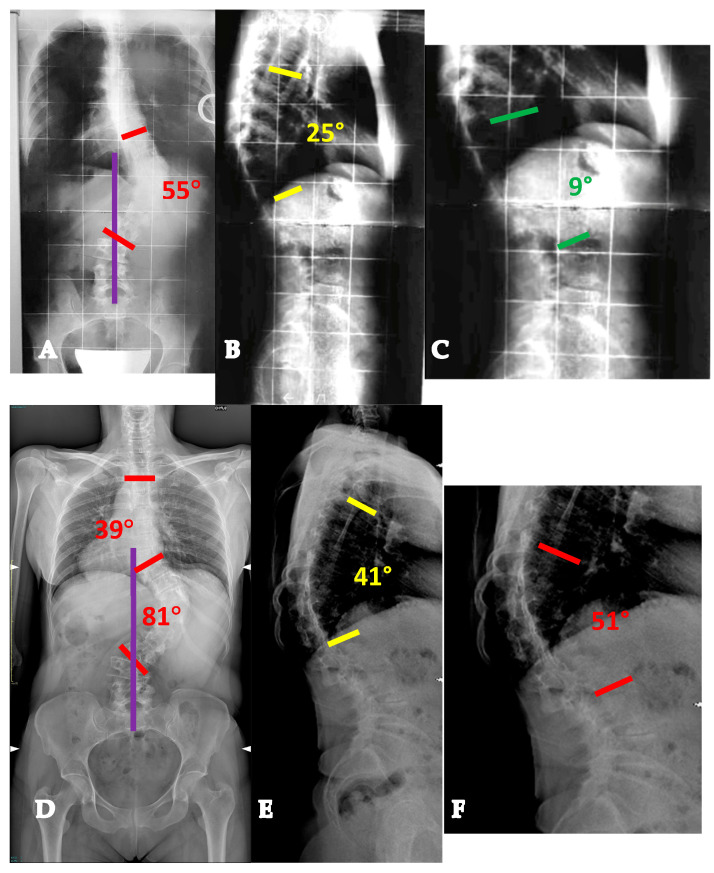
Fifty-five-year-old woman with an AdIS 2 curve. On top (**A**–**C**), radiographic appearance at 31 years old, before 2 pregnancies. A mild coronal curve with a good sagittal balance can be seen. Below (**D**–**F**), radiographic appearance at 55 years old, after two pregnancies and menopause. A progression of the coronal deformity can be seen, as well as a severe thoracolumbar junction kyphotic change.

**Figure 9 healthcare-13-02418-f009:**
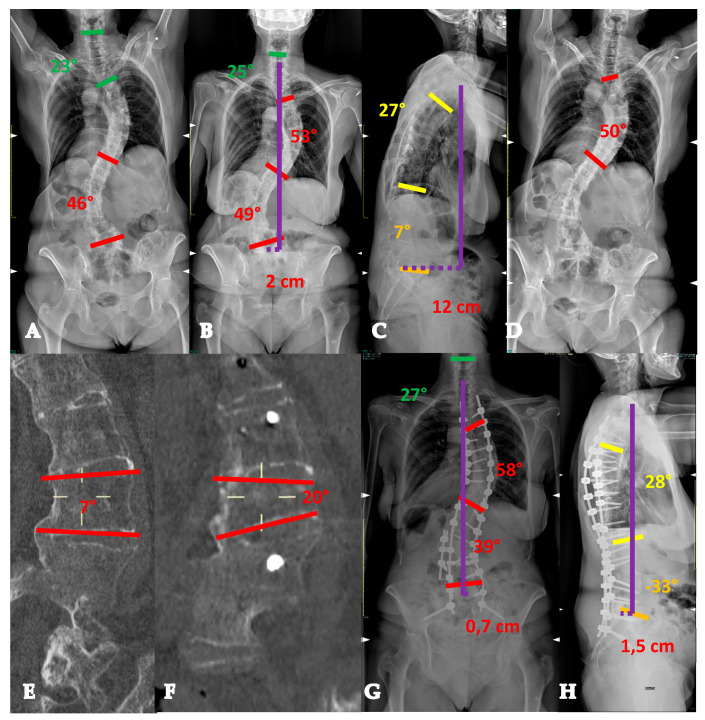
Seventy-one-year-old patient with an AdIS 2 curve. She has a severe multiplanar deformity, with a stiff coronal curve and a lumbar flat-back deformity with anterior imbalance (**A**–**D**). An L3 asymmetric PSO was planned. On the CT scan, the change in L3 body shape can be seen between preoperative (**E**) and postoperative (**F**). Surgery improved the overall coronal and sagittal balance (**G**,**H**).

**Figure 10 healthcare-13-02418-f010:**
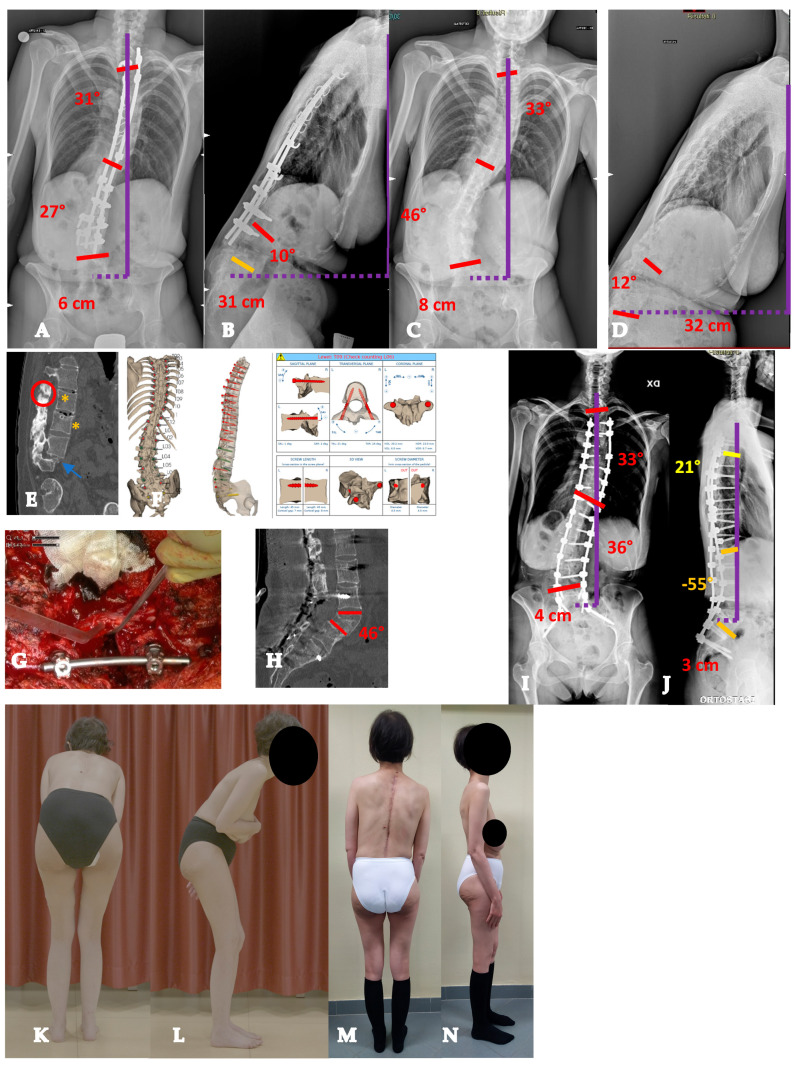
Thirty-nine-year-old patient with an AdIS 2 curve. She underwent prior posterior fusion with residual coronal and sagittal imbalance (**A**,**B**). The first surgical step required removal of the instrumentation (**C**,**D**) and CT scan acquisition. The CT scan (**E**) showed a fusion mass with several areas of pseudoarthrosis (circle and asterisk) and a stiff lumbosacral kyphosis (arrow). Patient-specific 3D-printed guides were manufactured based on the CT scan (**F**). L5 PSO was performed (**G**), intraoperative image; (**H**) postoperative CT scan showing the correction. Surgery allowed us to improve both coronal and sagittal balance (**I**,**J**). Clinically, a significant improvement can be seen from preoperative (**K**,**L**) to postoperative (**M**,**N**) appearance.

**Figure 11 healthcare-13-02418-f011:**
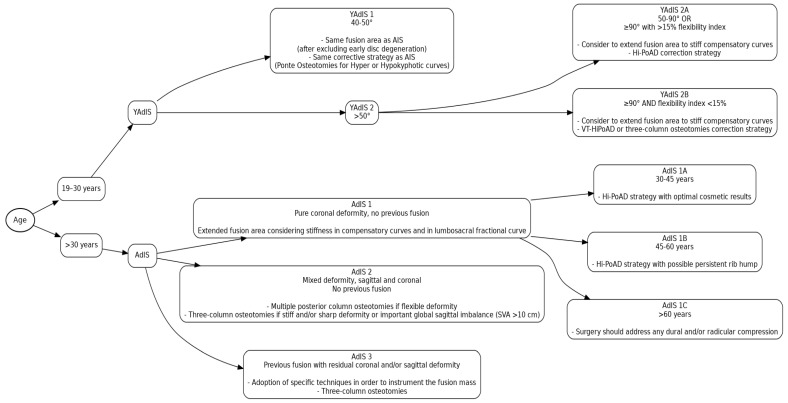
Flowchart of the proposed classification for adult idiopathic scoliosis (AdIS) and young adult idiopathic scoliosis (YAdIS), with corresponding surgical strategies. **Abbreviations:** AIS, Adolescent Idiopathic Scoliosis; Hi-PoAD, High-Power Asymmetric Derotation; VT-HiPoAD, Vertebral Translation + High-Power Asymmetric Derotation; SVA, Sagittal Vertical Axis.

## Data Availability

Data are available upon reasonable request.

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
