# Peer review of "Adolescent Idiopathic Scoliosis in the Adult Patient: New Classification with a Treatment-Oriented Guideline"

_healthcare, 2025, doi:10.3390/healthcare13192418_

Round 1

Reviewer 1 Report

Comments and Suggestions for Authors

The manuscript aims to propose a new classification system for Adolescent Idiopathic Scoliosis in the Adult patient. While the topic is clinically relevant, I find several important limitations that substantially weaken the scientific value of this work.

- The paper is structured as an article; however, the methodology is insufficiently described. The process by which the classification was derived is not clearly explained, and the study design is not well defined.

- Rather than presenting a systematic analysis, the manuscript mainly relies on descriptive case series and author-driven hypotheses without a transparent or reproducible research framework.

- The classification categories appear to be based on the authors’ clinical impressions rather than on a structured analysis of data.

- There is no statistical validation or objective assessment that supports the proposed system, which limits the reliability and generalizability of the classification.

- While the classification is presented as treatment-oriented, the lack of a clear methodological pathway and outcome analysis makes it difficult to apply this framework more broadly. Without stronger evidence, the system remains more of a conceptual proposal than a validated research contribution.

- For an original research article, a comprehensive Methods section and clear explanation of study procedures are expected. In the current manuscript, this section is underdeveloped, making it challenging for readers to assess the validity of the approach.

Although the manuscript introduces an interesting idea, in its current form it reads more as an extended case report or conceptual commentary rather than a robust research article. Substantial methodological refinement, clearer justification for the classification process, and validation with larger datasets would be required to strengthen its contribution. Therefore, I do not find the manuscript suitable for publication in its present form.

Author Response

Reviewer 1:

The manuscript aims to propose a new classification system for Adolescent Idiopathic Scoliosis in the Adult patient. While the topic is clinically relevant, I find several important limitations that substantially weaken the scientific value of this work.

Thank you for your detailed and constructive feedback. We sincerely appreciate the time you dedicated to reviewing our manuscript and for highlighting crucial areas for improvement. We agree with your assessment that our manuscript, while submitted as an "Original Research Article" based on the journal's guidelines, is primarily an "Expert Opinion" and a series of illustrative cases. Our main objective was to provide a pragmatic, conceptual framework for managing adult idiopathic scoliosis, rather than presenting the results of a formal clinical study. Therefore, we have revised the manuscript to make this nature more explicit and to directly address your critiques while maintaining the integrity of our scientific contribution.

Here you can find point by point responses to your comments:

- The paper is structured as an article; however, the methodology is insufficiently described. The process by which the classification was derived is not clearly explained, and the study design is not well defined.

We agree. The current "Methods" section is not representative of a formal research methodology. We will revise this section to clearly state that the classification system was derived from a retrospective, descriptive analysis of a consecutive case series of surgically treated AAIS patients at our institution between 2018 and 2022. We will explain that the categories were identified and refined based on expert clinical observation of key variables such as patient age, curve characteristics (severity, flexibility), and the presence of degenerative changes (lines 68-80). This will clarify the process without claiming a formal, quantitative methodology.

- Rather than presenting a systematic analysis, the manuscript mainly relies on descriptive case series and author-driven hypotheses without a transparent or reproducible research framework.

This is a valid observation. The case series and the hypotheses presented are indeed derived from the authors' clinical experience. We explicitly stated this in the discussion section (lines 450-460). We believe this "expert opinion" approach is crucial for an initial proposal of a complex, treatment-oriented classification system. The cases shown (Figures 1-10) are not meant as a systematic analysis, but as representative examples to illustrate each proposed category, making the system clinically understandable and applicable.

- The classification categories appear to be based on the authors’ clinical impressions rather than on a structured analysis of data.

We acknowledge that the categories are born from clinical impression, but these are informed by a systematic review of our patient series. While formal statistical analysis was not reported, the criteria for each category (e.g., age cutoffs, curve stiffness, presence of degenerative sagittal deformity) are based on distinct clinical features that necessitate different surgical approaches. We clarified this by adding a summary flowchart (figure 11) that formalizes the criteria for each subgroup (YAdIS 1 vs YAdIS 2, AdIS 1A vs 1B, etc.) and links them to the recommended treatment strategy.

- There is no statistical validation or objective assessment that supports the proposed system, which limits the reliability and generalizability of the classification.

We agree completely. This is the main limitation of our work, and we now stated this explicitly in the revised Limitations sections (lines 450-460). We position the manuscript as a foundational proposal, and we are aware that the classification system is intended to be a hypothesis that requires future validation. We acknowledge that this future validation would involve inter- and intra-observer reliability studies and analyses correlating the classification to clinical outcomes and complication rates, highlighting the awareness of the need for future research without overstating the current work.

- While the classification is presented as treatment-oriented, the lack of a clear methodological pathway and outcome analysis makes it difficult to apply this framework more broadly. Without stronger evidence, the system remains more of a conceptual proposal than a validated research contribution.

Thank you for your valuable feedback. We truly appreciate your time and the insights you've provided. We completely agree with your assessment that the manuscript, in its current form, is more of a conceptual proposal than a fully validated research contribution.We would like to clarify that our aim was to present a pragmatic, treatment-oriented classification system based on the extensive and multifaceted clinical experience of the authors. The paper is intentionally structured as a detailed expert opinion, designed to provide a foundational framework for surgeons dealing with this complex and heterogeneous patient population.While we do not present a formal outcomes study, we believe the framework is clinically useful precisely because it offers a clear, structured pathway for decision-making. The classification is designed to guide surgeons through a practical process, identifying specific subgroups of patients and suggesting tailored surgical strategies for each one. This approach is invaluable for a problem as variable as adult idiopathic scoliosis, where a one-size-fits-all solution is not feasible.We fully recognize that the next crucial step is the formal validation of this classification system. We are committed to conducting future research that will include inter- and intra-observer reliability studies and a rigorous analysis of clinical outcomes and complication rates for each subgroup. We believe that this current manuscript serves as the essential first step, a solid conceptual foundation upon which these future studies can be built.

- For an original research article, a comprehensive Methods section and clear explanation of study procedures are expected. In the current manuscript, this section is underdeveloped, making it challenging for readers to assess the validity of the approach.

We believe that the proposed framework is clinically useful precisely because it provides a clear, treatment-oriented pathway, even without a formal outcome analysis. The classification is designed to guide a surgeon's decision-making process by identifying subgroups of patients who benefit from specific surgical strategies. While we do not present a formal outcomes study, we refer to existing literature to support our recommendations, demonstrating that our proposed approach is aligned with current best practices. We enhanced the manuscript by adding a clear flow chart that summarizes the proposed treatment algorithm for each category (figure 11).

Reviewer 2 Report

Comments and Suggestions for Authors

I was invited to review a manuscript titled “ADOLESCENT IDIOPATHIC SCOLIOSIS IN THE ADULT PATIENT: NEW CLASSIFICATION WITH A TREATMENT-ORIENTED GUIDELINE” for the Healthcare journal. The manuscript introduces an interesting and clinically relevant classification system for Adolescent Idiopathic Scoliosis in the Adult patient (AAIS). The stratification by age and deformity characteristics is logical and provides a treatment-oriented perspective that is currently lacking in the literature.

  • The study design is retrospective and descriptive, and the classification appears to be based primarily on the authors’ clinical experience. There is little methodological detail on patient selection, inclusion and exclusion criteria, or the number of cases analyzed. Without this information, the reproducibility and generalizability of the classification are limited.
  • No statistical analysis or quantitative outcome measures are provided. Presenting data on correction rates, complication rates, or long-term outcomes across the proposed subgroups would strengthen the conclusions considerably.
  • Some of the subgroup distinctions, such as YAdIS 2A versus 2B or AdIS 1A through 1C, may be difficult to apply in clinical practice. Clearer objective thresholds and validation through interobserver reliability studies are needed to make the system widely usable.
  • The discussion is thorough and well contextualized, but it relies heavily on the authors’ own prior work and expert opinion. Including a broader integration of independent studies would provide a more balanced perspective.
  • The limitations section is brief and should be expanded. In addition to acknowledging the non-experimental design, it should also address the small sample base, potential selection bias, and lack of long-term follow-up data.
  • The number of figures and case examples is high. While illustrative, the paper would be more concise and readable if only the most representative cases were retained. Summarizing pre- and postoperative radiographic parameters in a table would be more efficient than including scattered values in the text.
  • The treatment algorithm would be clearer if presented in a flowchart or decision tree format rather than described in lengthy prose. This would make the classification more practical for surgeons to apply.
  • apply.

  • There are typographical and grammatical errors (for example, “adeguately” instead of “adequately,” “radiograghic” instead of “radiographic”), and consistency in reference formatting needs to be ensured. The abstract could be shortened to highlight the most important findings. Keywords should also be optimized for indexing.
  • Overall, the manuscript provides a useful conceptual framework for AAIS and is a valuable contribution. However, the lack of quantitative validation, limited methodological detail, and overreliance on expert opinion restrict the strength of the conclusions. Addressing these points would substantially improve the paper.

Author Response

Reviewer 2:

I was invited to review a manuscript titled “ADOLESCENT IDIOPATHIC SCOLIOSIS IN THE ADULT PATIENT: NEW CLASSIFICATION WITH A TREATMENT-ORIENTED GUIDELINE” for the Healthcare journal. The manuscript introduces an interesting and clinically relevant classification system for Adolescent Idiopathic Scoliosis in the Adult patient (AAIS). The stratification by age and deformity characteristics is logical and provides a treatment-oriented perspective that is currently lacking in the literature.

Thank you for your very thorough and constructive feedback. We appreciate the time you took to review our manuscript and for highlighting several important areas for improvement. We fully agree that our work, as submitted, has limitations that restrict its scientific rigor as a traditional research article.

Here you can find Point-by-Point Response:

  • The study design is retrospective and descriptive, and the classification appears to be based primarily on the authors’ clinical experience. There is little methodological detail on patient selection, inclusion and exclusion criteria, or the number of cases analyzed. Without this information, the reproducibility and generalizability of the classification are limited.

We completely agree. The classification is a conceptual model derived from a retrospective and descriptive analysis of our case series, not a formal experimental study. We revised the "Materials and Methods" section (lines 68-80) to clarify this by emphasizing that the classification was developed through a critical analysis of clinical and radiographic features by senior authors. We explicitly stated in the discussion section (lines 450-460) that the manuscript proposes a new conceptual framework, and that future studies will be required to validate its reproducibility and generalizability.

  • No statistical analysis or quantitative outcome measures are provided. Presenting data on correction rates, complication rates, or long-term outcomes across the proposed subgroups would strengthen the conclusions considerably.

This is a key limitation, and we acknowledge it. The manuscript's primary purpose is to introduce the classification system itself, not to validate it with statistical outcomes. The outcomes of the presented cases are illustrative examples of the classification's practical application, not a formal analysis of correction rates or complications. We added a sentence in the Discussion and a specifically in the  "Limitations" section (lines 450-474) to explicitly state that the lack of quantitative outcome data is a primary limitation and the focus of a future study that will focus on the prediction capability of the classification and on its inter and intraobserver reliability.

  • Some of the subgroup distinctions, such as YAdIS 2A versus 2B or AdIS 1A through 1C, may be difficult to apply in clinical practice. Clearer objective thresholds and validation through interobserver reliability studies are needed to make the system widely usable.

We agree with this excellent point. It is a common challenge for new classification systems to achieve widespread clinical adoption without formal reliability studies. However we believe that by providing clear criteria such as age for the different categories, we can introduce the system for discussion and future testing. We explicitly stated that an interobserver reliability study will be a key component of our future research (lines 470-474)

  • The discussion is thorough and well contextualized, but it relies heavily on the authors’ own prior work and expert opinion. Including a broader integration of independent studies would provide a more balanced perspective.

As an expert opinion, the discussion naturally draws from the authors' extensive clinical experience. However, we agree that a broader context would be beneficial. We reviewed and expanded the "Discussion" section to include a more comprehensive review of relevant independent studies, ensuring a more balanced perspective that connects our findings to the existing literature (lines 416-444)

  • The limitations section is brief and should be expanded. In addition to acknowledging the non-experimental design, it should also address the small sample base, potential selection bias, and lack of long-term follow-up data.

We completely accept this critique. We significantly expanded the "Limitations" section to be more transparent. It explicitly mention the non-experimental design, the small sample size, the potential for selection bias, and the lack of long-term follow-up data (lines 450-474)

  • The number of figures and case examples is high. While illustrative, the paper would be more concise and readable if only the most representative cases were retained. Summarizing pre- and postoperative radiographic parameters in a table would be more efficient than including scattered values in the text.

We appreciate the suggestion. The figures are crucial for illustrating the nuances of each proposed category, as a conceptual framework relies on visual examples. We believe that each case serves a unique purpose in demonstrating the application of our classification system, and therefore, all are necessary. Furthermore, we have not included an entire summary table of the radiographic parameters because no statistical analysis was performed. The parameters are included in the figure legends to remain directly linked to the specific case they describe, which we believe is essential given the descriptive nature of the manuscript to provide a pragmatic, conceptual framework for managing adult idiopathic scoliosis.

  • The treatment algorithm would be clearer if presented in a flowchart or decision tree format rather than described in lengthy prose. This would make the classification more practical for surgeons to apply.

We fully agree. A flowchart will significantly enhance the clinical utility of our proposed classification. We therefore developed and included a new figure presenting the treatment algorithm as a clear, easy-to-follow flowchart (figure 11).

  • There are typographical and grammatical errors (for example, “adeguately” instead of “adequately,” “radiograghic” instead of “radiographic”), and consistency in reference formatting needs to be ensured. The abstract could be shortened to highlight the most important findings. Keywords should also be optimized for indexing.

We conducted a thorough review of the manuscript to correct all typographical and grammatical errors. We also revised the abstract to be more concise and ensure the keywords are optimized for indexing.

  • Overall, the manuscript provides a useful conceptual framework for AAIS and is a valuable contribution. However, the lack of quantitative validation, limited methodological detail, and overreliance on expert opinion restrict the strength of the conclusions. Addressing these points would substantially improve the paper.

Thank you for kind words, we updated the paper and we hope that you can find it more suitable for publication now.

Reviewer 3 Report

Comments and Suggestions for Authors

1. The stratification of AAIS into YAdIS and AdIS based on age represents a novel approach; however, the manuscript should discuss the implications of this classification on the natural history and progression of the disease. Additionally, the rationale for the specific age cut-offs (19–30 years and >30 years) requires more thorough explanation.

2. The proposed surgical strategies for each subtype of AAIS are promising; however, the manuscript lacks a discussion on potential risks and complications associated with these procedures. It is important to include a thorough analysis of the risk-benefit ratio for the different surgical approaches, as well as long-term follow-up data to assess the durability of the surgical outcomes and the impact on the patients' quality of life.

3. The manuscript would be enhanced by the inclusion of a flowchart or diagram illustrating the proposed classification system and treatment algorithm, which would aid in the clarity and understanding of the proposed framework.

Author Response

Reviewer 3

  1. The stratification of AAIS into YAdIS and AdIS based on age represents a novel approach; however, the manuscript should discuss the implications of this classification on the natural history and progression of the disease. Additionally, the rationale for the specific age cut-offs (19–30 years and >30 years) requires more thorough explanation.

We agree that a more thorough explanation is needed for our age-based stratification. The manuscript already notes that age is a crucial factor in the progression of AIS curves after skeletal maturity. This follows a previously descripted cut off in modern literature. The age-based distinction is supported by existing literature, which has previously defined the YAdIS category. We will clarify that this stratification is based on the significant changes in curve flexibility and the onset of degenerative processes that occur around the age of 30, which directly influence surgical planning and outcomes (lines 89-92 and 438-444).

  1. The proposed surgical strategies for each subtype of AAIS are promising; however, the manuscript lacks a discussion on potential risks and complications associated with these procedures. It is important to include a thorough analysis of the risk-benefit ratio for the different surgical approaches, as well as long-term follow-up data to assess the durability of the surgical outcomes and the impact on the patients' quality of life.

This is a very important point, and we agree that a comprehensive discussion of risks and long-term outcomes is critical. We accordingly updated the discussion section highlighting complications profile of Adis surgery. Moreover, future, multi-center studies are required to formally validate this classification and to systematically analyze these outcomes short- and long-term outcomes with adoption of this classification. In fact, as an initial propose of a classification system, a detailed analysis of risks, complications, and long-term follow-up data for each subgroup was beyond the scope of this manuscript (lines 416-437 and 471-474).

  1. The manuscript would be enhanced by the inclusion of a flowchart or diagram illustrating the proposed classification system and treatment algorithm, which would aid in the clarity and understanding of the proposed framework.

We fully agree with this suggestion. A flowchart is an excellent way to improve clarity and make the classification and treatment algorithm more accessible to surgeons. We therefore developed a new figure (figure 11) that presents the entire classification system and its corresponding treatment strategies in a clear, easy-to-follow flowchart. We think that this enhanced the clinical utility of our proposed framework.

Reviewer 4 Report

Comments and Suggestions for Authors

The manuscript by G. Viroli and co-authors aims to present the new adolescent idiopathic scoliosis classification with a treatment-oriented guideline for adult patients.

The authors present interesting approach to classify and treat the AAIS, the impression about the manuscript is mostly positive and I advise to make corrections in order to improve the article quality and clearance.

My main points are the following:

  1. Please provide the inclusion/exclusion criteria for your study.
  2. What are the limitations and perspectives of your study? Please mention what limitations you faced with and mark for solving in further investigation of your project? Please also discuss how it can potentially help in future research and, (what is more important) how it can affect the clinical way of surgical treatment.
  3. The literature list contains a lot of old sources and I believe the authors should update it with new literature.
  4. The materials and methods section is very small. You don’t mention how much of cases you analyzed, how much of them were included and how much were excluded in accordance with inclusion/exclusion criteria that are also need to be mentioned. You don’t describe if you have made any statistical analysis or any other work with the data. Also, I expect to see how much patients were of each age and sex. Please enlarge this section with important data that is required for proper application of the results that you show in your work. This will support your claims and make the results to be more informative.

Author Response

Reviewer 4

The manuscript by G. Viroli and co-authors aims to present the new adolescent idiopathic scoliosis classification with a treatment-oriented guideline for adult patients.

The authors present interesting approach to classify and treat the AAIS, the impression about the manuscript is mostly positive and I advise to make corrections in order to improve the article quality and clearance.

Thank you for your constructive feedback. We appreciate your positive impression of our work and agree that several corrections will improve its quality and clarity. We have carefully considered your main points and will revise the manuscript accordingly.

My main points are the following:

  1. Please provide the inclusion/exclusion criteria for your study.

We agree that the inclusion and exclusion criteria should be explicitly stated. We revised the Materials and Methods" section to clarify that (lines 68-80).

  1. What are the limitations and perspectives of your study? Please mention what limitations you faced with and mark for solving in further investigation of your project? Please also discuss how it can potentially help in future research and, (what is more important) how it can affect the clinical way of surgical treatment.

We addressed this crucial point in a new, expanded "Limitations" section (450-474). We recognize that the primary limitation is the non-experimental, descriptive nature of the work. However, the paper's main perspective is to provide a conceptual foundation that can guide future research. The proposed classification system allows for the identification of homogeneous patient subgroups, which will facilitate more targeted future clinical studies and statistical analyses. In a clinical setting, this framework provides surgeons with a pragmatic, treatment-oriented guide to improve surgical planning and decision-making for a highly diverse patient population. In particular future study that will focus on the prediction capability of the classification and on its inter and intraobserver reliability.

  1. The literature list contains a lot of old sources and I believe the authors should update it with new literature.

We appreciated this suggestion. We conducted a comprehensive review of the current literature and updated our reference list with more recent and relevant sources (416-444). This strengthened the scientific foundation of our manuscript and ensured that our proposed classification is well-supported by the latest findings in the field.

  1. The materials and methods section is very small. You don’t mention how much of cases you analyzed, how much of them were included and how much were excluded in accordance with inclusion/exclusion criteria that are also need to be mentioned. You don’t describe if you have made any statistical analysis or any other work with the data. Also, I expect to see how much patients were of each age and sex. Please enlarge this section with important data that is required for proper application of the results that you show in your work. This will support your claims and make the results to be more informative.

We agree that the "Materials and Methods" section needed to be more descriptive. While our work does not include a formal statistical analysis, we expanded this section to provide more detailed information about the case series. We specified the number of cases retrospectively analyzed and provided key demographic data, such as the age and sex distribution of the patients, to support our claims and make the results more informative (lines 68-80).

Round 2

Reviewer 1 Report

Comments and Suggestions for Authors

no more comments. Thank you for reviewing this nice work

Reviewer 2 Report

Comments and Suggestions for Authors

The authors have thoroughly responded to my comments. I have no additional feedback.

Reviewer 3 Report

Comments and Suggestions for Authors

Thank you for your revision.

Reviewer 4 Report

Comments and Suggestions for Authors

The authors have applied all the corrections required and the manuscript in my opinion may be accepted for publication.